

# Environmental DNA detection and quantification of invasive red-eared sliders, *Trachemy scripta elegans*, in ponds and the influence of water quality

Aozora Kakuda[1], Hideyuki Doi[2], Rio Souma[3], Mariko Nagano[2], Toshifumi Minamoto[4] and Izumi Katano[1]

[1] Graduate School of Humanities and Sciences, Nara Women's University, Nara, Japan
[2] Graduate School of Simulation Studies, University of Hyogo, Kobe, Japan
[3] IDEA Consultants Inc., Yaizu, Japan
[4] Graduate School of Human Development and Environment, Kobe University, Kobe, Japan

## ABSTRACT

Environmental DNA (eDNA) is a powerful tool for monitoring the distribution of aquatic macro-organisms. However, environmental factors, including the water temperature and water quality, can affect the inhibition and/or degradation of eDNA, which complicates accurate estimations of eDNA concentrations and the detection of the presence/absence of species in natural habitats. Further very few eDNA studies have been conducted for reptiles, especially with respect to estimating their biomass and/or abundances. Here we examined the relationship between the visually-observed number of red-eared sliders (*Trachemys scripta elegans*) and eDNA concentrations across 100 ponds. Additionally, we evaluated the effect of water quality on red-eared slider eDNA concentration in these ponds. We found that there was a significant positive correlation between the observed number of red-eared sliders and the eDNA concentration in the ponds. On comparing various water quality indicators, including dissolved nitrogen, dissolved phosphorous, organic matter, and chlorophyll a (Chl. *a*), we found that only Chl. *a* had a negative correlation with the red-eared slider eDNA concentration, while we did not find any inhibition in the quantitative PCR. We conclude that concentrations of eDNA can potentially be used for estimating the abundance of the red-eared slider. Additionally, Chl. *a* might indirectly influence the degradation of eDNA through the microorganisms bonded to the phytoplankton in the ponds, as microbial activity is thought to decrease eDNA persistence.

## INTRODUCTION

Environmental DNA (eDNA) methods for monitoring the distribution of aquatic species have recently been developed (*Ficetola et al., 2008*; *Rees et al., 2014*; *Goldburg, Strickler & Pilliod, 2015*). The eDNA consists of DNA fragments released through the mucus, urine, gametes, or feces of species in the environment. In 2008, *Ficetola et al. (2008)* first used the eDNA method for a macro-organism, the American bullfrog (*Rana catesbeiana*),

Corresponding authors
Hideyuki Doi,
hideyuki.doi@icloud.com
Izumi Katano,
katano@cc.nara-wu.ac.jp,
katanon2003@yahoo.co.jp

to detect their presence in ponds using water samples. It is possible to analyze DNA fragments of target species from a few liters or less of water (*Ficetola et al., 2008*; *Rees et al., 2014*; Goldberg & Strickler, 2015; *Deiner et al., 2017*). As we just sample the water in the field, eDNA methods are non-invasive, take a shorter time, and cost less compared with traditional monitoring methods, capture or visual survey (*Thomsen & Willerslev, 2015*).

The eDNA analysis has been applied to various aquatic taxa, for example, fish (*Minamoto et al., 2012*; *Thomsen et al., 2012a*; *Takahara et al., 2012*; *Takahara, Minamoto & Doi, 2013*; *Eichmiller, Miller & Sorensen, 2016a*), amphibians (*Ficetola et al., 2008*; *Pilliod et al., 2013*; *Fukumoto, Ushimaru & Minamoto, 2015*; *Katano et al., 2017*), mollusks (*Goldberg et al., 2013*), crustaceans (*Tréguier et al., 2014*), insects (*Thomsen et al., 2012b*; *Doi et al., 2017b*), trematodes (*Huver et al., 2015*; *Hashizume et al., 2017*), and aquatic plants (*Fujiwara et al., 2016*). However, there are few studies for reptiles, especially on the eDNA quantification of turtles (e.g., *Davy, Kidd & Wilson, 2015*; *De Souza et al., 2016*; *Lacoursière-Roussel et al., 2016*).

Here, we conducted research to determine the distribution and abundance of a turtle species, the red-eared slider (*Trachemys scripta elegans*). The red-eared slider is listed in the top 100 of the world's worst invasive species by the World Conservation Union IUCN (Global Invasive Species Database, http://www.issg.org/database). In Japan, red-eared sliders were imported from USA as a pet in the 1950s, and since then it has been released into the local natural habitats (*Oi et al., 2011*). This species is an omnivore and it has gained some attention as they affect indigenous animals (*Lever, 2003*). In 2013, The Nature Conservation Society of Japan (NACS-J) conducted a visual survey in 41 of the 47 prefectures in Japan ($N = 4,146$), and reported that 64% of the turtle individuals observed were identified as red-eared sliders (*NACS-J, 2013*). Red-eared sliders disrupt pond ecosystems and have expanded their distribution into numerous ponds in Japan (*Taniguchi et al., 2017*). It is important to rapidly elucidate the distribution of invasive species to conserve the ecosystems in which they reside (*Pyšek & Richardson, 2010*); thus, methods to easily detect red-eared sliders, which are one of the common invasive species in Japan, should be developed. However, the effective sampling methods for turtles depends on the target species. *Sterrett et al. (2010)*, suggested that we might underestimate the abundance of red-eared sliders by using visual observations. Thus, eDNA methods would be a useful tool to replace visual observations for evaluating the turtle distribution.

Despite the merits of eDNA methods for turtle surveying, previous eDNA studies in freshwater systems suggest that a number of environmental factors affect the probability of eDNA detection; for example, the water temperature and water quality, including the pH, suspended solid (SS), total phosphorous (TP), total nitrogen (TN), biological oxygen demand (BOD), and chlorophyll a (Chl. *a*) (e.g., *Barnes et al., 2014*; *Strickler, Fremier & Goldberg, 2014*; *Eichmiller, Best & Sorensen, 2016b*; *Song, Small & Casman, 2017*). These factors can lead to false negative detections, which prevents the accurate evaluation of eDNA concentrations and detection of the distribution/quantification of a species. To improve the eDNA evaluation of distribution and biomass/abundance in natural habitats, we should understand the relationship between eDNA detection rate/concentration and these environmental factors.

Two main environmental factors that negatively influence eDNA detection rate are 'inhibition' and 'degradation'. In fact, humic acids are known to inhibit DNA polymerase in PCR (*Matheson et al., 2010*), and therefore, it could directly impact on eDNA detection. Mesocosm studies have found that a number of factors are important to decreasing the degradation of eDNA, such as low water temperatures, low UV-B levels, alkaline (high-pH), and increased BOD, increased Chl. *a* concentration, and increased total eDNA concentration (*Barnes et al., 2014*; *Strickler, Fremier & Goldberg, 2014*). Although these studies are based on laboratory experiments, in rivers, eDNA detection rate has been known to decrease with increasing Chl. *a*, but increase with increasing water temperature and pH (*Song, Small & Casman, 2017*). Because the effect of Chl. *a* was different in laboratory vs. in situ experiments, more evidence is needed to understand the effects of water quality on eDNA degradation; however, there are few studies on this conducted in the field.

We hypothesized that eDNA would be a suitable method to determine the distribution of red-eared sliders and that water quality would influence/inhibit eDNA measurement by quantitative real-time PCR (qPCR). In this study, our aim was to compare the eDNA concentrations of the target species, by using qPCR paired with visual observations of the turtles. Specifically, we evaluated the species abundance with the measurements of eDNA concentrations in 100 study ponds. From the eDNA concentration and water quality data, we examined the relationships between the water quality and eDNA concentrations of the red-eared slider, to consider the water quality effects that influence eDNA degradation and PCR inhibition in the ponds.

## MATERIALS & METHODS

### Study site

We conducted the field survey in 100 ponds that were located in Himeji, Japan (34°47′–34°54′N, 134°35′–134°45′E, Fig. 1) between July 21 and November 16, 2016. We randomly selected ponds located in each area category, comprising city, rural, and mountain areas (see Tables 1A, 1B, Fig. S1 in SEM). There are few ponds in the southern (city area) and northern areas (mountain) because the distribution of the ponds is biased. We conducted statistical analysis among the three area categories and eDNA concentration, but did not find remarkable patterns of eDNA concentration and detection among the pond locations. The field survey and pond sampling was permitted by the land owners, if needed.

### Field survey and sampling

We recorded the presence/absence and number of red-eared sliders, based on visual observations from the shore line for three minutes by an expert (A. Kakuda). We performed surveys during the daytime (1000–1300 h) each day, except for rainy days and one day after rain. From a point within each pond, 500 mL of surface water was collected for eDNA and SS analysis, and a further 100 mL of surface water was collected for Chl. *a* and water quality analysis. In this study, we conducted a field survey to compare the convenience of the eDNA method with visual surveys; thus, we decided to increase the number of sampling ponds rather than increase the replications at each pond. We selected a water sampling point in the middle section of the ponds, far from the water outflows/inflows.

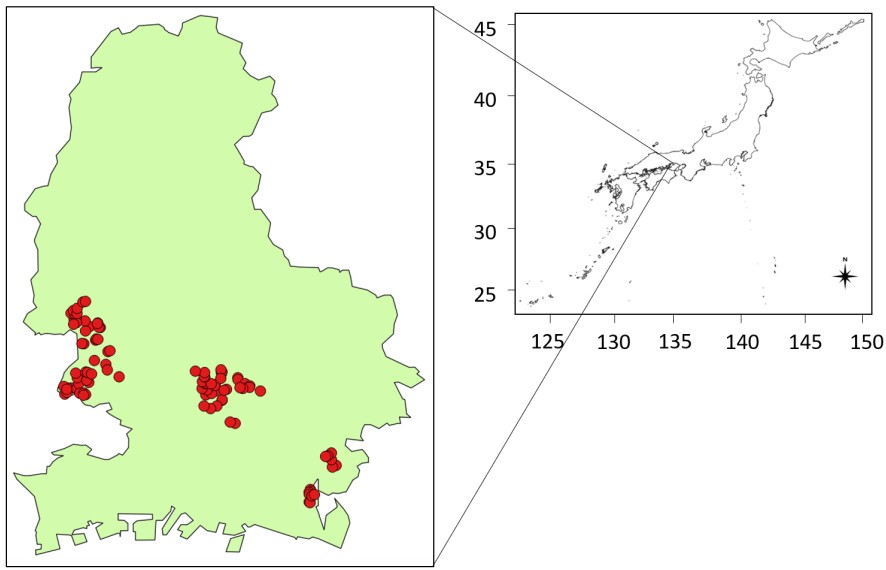

**Figure 1** **Study sites represented by red points.**

We directly sampled the eDNA using a bleached bottle and added 0.5 mL of benzalkonium chloride (BAC) to avoid a reduction of the eDNA concentration in the samples (*Yamanaka et al., 2016*). Samples were stored in a cooler box with a 'cooler blank'. The 'cooler blank' contained 500 mL of DNA-free water, which we brought to the field, and it was treated identically to the other water samples, except that it was not opened at the field sites.

## Water preparation

Within six hours after water sampling, the samples were filtered onto a GF/F glass filter (47 mm diameter and 0.7 μm pore size, GE Healthcare Japan, Tokyo, Japan). We used separate filters for the eDNA, SS (from 500 mL water), and Chl. *a* (from 100 mL water) analysis. The filter was then wrapped in commercial aluminum foil and stored at −20 °C until eDNA extraction, or SS/ Chl. *a* measurement. For eDNA samples, the 'cooler blank' and a 'filter blank' consisting of DNA-free distilled water were filtered in the same way as the samples. To avoid contamination, each piece of equipment that was used in the water sampling or filtration was soaked in a 10% commercial bleach solution (approximately 0.6% sodium hypochlorite) and rinsed using DNA-free distilled water prior to reuse. The 80 mL of the filtrated samples were stored at −20 °C until further water quality analyses. We lost some of the samples for water quality measurements (see the missing values of Tables S1A, S1B in SEM).

## DNA extraction from the filters

The eDNA was extracted from each filter using a DNeasy Blood & Tissue Kit (Qiagen, Hilden, Germany) based on the method described by *Uchii, Doi & Minamoto (2016)*. Each filtrate was soaked in 400 μL of buffer AL and 40 μL of protease K in a Salivette tube (Sarstedt, Nümbrecht, Germany) and incubated at 56 °C for 30 min. After centrifugation

**Table 1 Sampling date, location, the detection ofred-eared slidereDNA,and the number of red-eared slidersvisually observed in the study ponds.** Observed by both eDNA and visual observation is ◎. Observed by only eDNA and by only visual observation are ○ and ●, respectively.

**(A)**

| Pond No. | Date | Latitude | Longtitude | eDNA detected | Visual observation |
|---|---|---|---|---|---|
| 1 | 2016/7/21 | 34°51′27″71 | 134°40′36″11 | ◎ | 10 |
| 2 | 2016/7/29 | 34°51′15″84 | 134°41′12″84 | ● | 2 |
| 3 | 2016/7/29 | 34°51′16″92 | 134°41′12″84 | ○ | – |
| 4 | 2016/8/4 | 34°51′02″88 | 134°41′00″96 | – | – |
| 5 | 2016/8/4 | 34°50′57″12 | 134°40′46″92 | ◎ | 1 |
| 6 | 2016/8/8 | 34°51′36″36 | 134°40′57″57 | ◎ | 2 |
| 7 | 2016/8/8 | 34°51′30″96 | 134°40′49″08 | ○ | – |
| 8 | 2016/8/10 | 34°51′02″88 | 134°40′32″16 | – | – |
| 9 | 2016/8/10 | 34°51′50″27 | 134°40′36″26 | – | – |
| 10 | 2016/8/16 | 34°51′38″88 | 134°40′39″39 | ◎ | 10 |
| 11 | 2016/8/16 | 34°51′42″12 | 134°40′27″84 | ● | 1 |
| 12 | 2016/8/19 | 34°50′24″24 | 134°41′42″42 | – | – |
| 13 | 2016/8/19 | 34°50′26″88 | 134°41′30″84 | ◎ | 3 |
| 14 | 2016/8/26 | 34°51′52″92 | 134°40′35″04 | ◎ | 1 |
| 15 | 2016/10/7 | 34°52′15″97 | 134°41′11″45 | – | – |
| 16 | 2016/10/7 | 34°52′21″91 | 134°41′10″54 | ◎ | 1 |
| 17 | 2016/10/7 | 34°52′19″39 | 134°41′12″48 | – | – |
| 18 | 2016/10/7 | 34°52′24″05 | 134°41′10″23 | – | – |
| 19 | 2016/10/7 | 34°52′06″21 | 134°41′48″00 | ● | 1 |
| 20 | 2016/10/7 | 34°52′02″02 | 134°41′48″27 | ○ | – |
| 21 | 2016/10/14 | 34°51′36″40 | 134°42′38″65 | – | – |
| 22 | 2016/10/14 | 34°51′52″94 | 134°42′04″27 | – | – |
| 23 | 2016/10/14 | 34°51′47″99 | 134°42′14″86 | ○ | – |
| 24 | 2016/10/14 | 34°51′44″67 | 134°42′14″22 | – | – |
| 25 | 2016/10/14 | 34°51′42″18 | 134°41′59″00 | – | – |
| 26 | 2016/10/14 | 34°51′44″11 | 134°41′52″91 | – | – |
| 27 | 2016/10/14 | 34°51′40″18 | 134°41′22″60 | – | – |
| 28 | 2016/10/14 | 34°51′38″16 | 134°41′17″16 | ● | 1 |
| 29 | 2016/10/14 | 34°51′50″62 | 134°40′57″02 | – | – |
| 30 | 2016/10/14 | 34°51′48″96 | 134°40′56″23 | ● | 3 |
| 31 | 2016/10/17 | 34°51′54″45 | 134°40′37″75 | – | – |
| 32 | 2016/10/17 | 34°51′57″51 | 134°40′28″63 | – | – |
| 33 | 2016/10/17 | 34°51′52″84 | 134°40′48″37 | – | – |
| 34 | 2016/10/19 | 34°51′43″21 | 134°35′35″21 | ○ | – |
| 35 | 2016/10/19 | 34°51′43″34 | 134°35′26″03 | ● | 1 |
| 36 | 2016/10/19 | 34°51′46″64 | 134°35′17″99 | – | – |
| 37 | 2016/10/19 | 34°51′41″91 | 134°35′46″99 | – | – |

**(A)**

| Pond No. | Date | Latitude | Longtitude | eDNA detected | Visual observation |
|---|---|---|---|---|---|
| 38 | 2016/10/19 | 34°51′53″15 | 134°35′51″31 | – | – |
| 39 | 2016/10/19 | 34°52′06″27 | 134°35′47″93 | – | – |
| 40 | 2016/10/19 | 34°52′16″58 | 134°35′44″54 | – | – |
| 41 | 2016/10/19 | 34°51′31″47 | 134°35′53″56 | ◉ | 4 |
| 42 | 2016/10/19 | 34°51′32″04 | 134°36′02″88 | ◉ | 1 |
| 43 | 2016/10/19 | 34°51′28″49 | 134°36′06″63 | ● | 5 |
| 44 | 2016/10/19 | 34°51′27″27 | 134°36′02″16 | – | – |
| 45 | 2016/10/19 | 34°51′29″60 | 134°35′20″10 | – | – |
| 46 | 2016/10/19 | 34°51′33″48 | 134°35′25″09 | ○ | – |
| 47 | 2016/10/19 | 34°51′40″47 | 134°35′24″34 | – | – |
| 48 | 2016/10/21 | 34°51′55″82 | 134°36′13″86 | – | – |
| 49 | 2016/10/21 | 34°52′00″86 | 134°36′09″34 | – | – |
| 50 | 2016/10/21 | 34°52′19″20 | 134°36′07″22 | – | – |

**(B)**

| Pond No. | Date | Latitude | Longtitude | eDNA detected | Visual observation |
|---|---|---|---|---|---|
| 51 | 2016/10/21 | 34°52′18″12 | 134°36′11″16 | ● | 3 |
| 52 | 2016/10/21 | 34°52′14″80 | 134°36′20″07 | – | – |
| 53 | 2016/10/21 | 34°52′36″15 | 134°36′52″11 | – | – |
| 54 | 2016/10/21 | 34°52′23″71 | 134°36′55″38 | – | – |
| 55 | 2016/10/21 | 34°52′08″35 | 134°37′21″82 | – | – |
| 56 | 2016/10/26 | 34°48′49″62 | 134°45′28″46 | – | – |
| 57 | 2016/10/26 | 34°48′45″99 | 134°45′20″11 | – | – |
| 58 | 2016/10/26 | 34°49′02″53 | 134°45′17″81 | – | – |
| 59 | 2016/10/26 | 34°49′17″61 | 134°45′17″76 | – | – |
| 60 | 2016/10/26 | 34°49′11″13 | 134°45′09″65 | ○ | – |
| 61 | 2016/10/26 | 34°49′09″75 | 134°45′04″08 | ○ | – |
| 62 | 2016/10/26 | 34°47′55″70 | 134°44′30″60 | – | – |
| 63 | 2016/10/26 | 34°47′51″66 | 134°44′32″38 | – | – |
| 64 | 2016/10/26 | 34°47′49″32 | 134°44′27″66 | – | – |
| 65 | 2016/10/26 | 34°47′44″73 | 134°44′28″25 | – | – |
| 66 | 2016/10/26 | 34°47′27″68 | 134°44′28″04 | – | – |
| 67 | 2016/10/26 | 34°47′26″17 | 134°44′30″00 | – | – |
| 68 | 2016/10/26 | 34°47′41″37 | 134°44′34″11 | – | – |
| 69 | 2016/10/26 | 34°47′44″19 | 134°44′39″70 | – | – |
| 70 | 2016/11/2 | 34°52′04″71 | 134°41′09″82 | ○ | – |
| 71 | 2016/11/2 | 34°52′08″04 | 134°40′33″96 | ◉ | 40 |
| 72 | 2016/11/2 | 34°52′08″77 | 134°40′32″74 | ◉ | 10 |
| 73 | 2016/11/2 | 34°52′17″98 | 134°40′33″59 | ● | 1 |
| 74 | 2016/11/2 | 34°52′20″97 | 134°40′12″81 | – | – |
| 75 | 2016/11/10 | 34°52′44″93 | 134°36′26″46 | – | – |
| 76 | 2016/11/10 | 34°53′04″92 | 134°36′56″88 | – | – |
| 77 | 2016/11/10 | 34°53′06″74 | 134°37′01″27 | – | – |
| 78 | 2016/11/10 | 34°53′22″42 | 134°36′02″79 | – | – |

**Table 1** (*continued*)

**(B)**

| Pond No. | Date | Latitude | Longtitude | eDNA detected | Visual observation |
|---|---|---|---|---|---|
| 79 | 2016/11/10 | 34°53′22″97 | 134°35′58″44 | – | – |
| 80 | 2016/11/10 | 34°53′30″58 | 134°36′29″18 | – | – |
| 81 | 2016/11/10 | 34°53′33″12 | 134°36′30″51 | – | – |
| 82 | 2016/11/10 | 34°53′32″29 | 134°36′34″27 | – | – |
| 83 | 2016/11/10 | 34°54′00″96 | 134°36′15″60 | – | – |
| 84 | 2016/11/10 | 34°53′51″23 | 134°36′06″41 | – | – |
| 85 | 2016/11/16 | 34°53′58″77 | 134°36′39″56 | – | – |
| 86 | 2016/11/16 | 34°53′59″25 | 134°36′36″64 | – | – |
| 87 | 2016/11/16 | 34°54′09″90 | 134°36′33″84 | – | – |
| 88 | 2016/11/16 | 34°54′07″92 | 134°36′33″12 | – | – |
| 89 | 2016/11/16 | 34°54′13″64 | 134°36′05″48 | – | – |
| 90 | 2016/11/16 | 34°54′19″24 | 134°35′45″65 | – | – |
| 91 | 2016/11/16 | 34°54′17″99 | 134°35′40″68 | – | – |
| 92 | 2016/11/16 | 34°54′09″87 | 134°35′46″19 | – | – |
| 93 | 2016/11/16 | 34°54′06″24 | 134°35′39″84 | – | – |
| 94 | 2016/11/16 | 34°54′30″99 | 134°35′33″98 | – | – |
| 95 | 2016/11/16 | 34°54′30″37 | 134°35′41″57 | – | – |
| 96 | 2016/11/16 | 34°54′36″81 | 134°35′39″32 | – | – |
| 97 | 2016/11/16 | 34°54′30″37 | 134°35′47″51 | – | – |
| 98 | 2016/11/16 | 34°54′41″97 | 134°35′47″75 | ● | 1 |
| 99 | 2016/11/16 | 34°54′56″00 | 134°36′00″70 | – | – |
| 100 | 2016/11/16 | 34°54′57″30 | 134°36′06″47 | – | – |

at 5,000 g for 5 min, 220 μL of TE buffer (pH 8.0) was added to each filter, and the tubes were centrifuged again in the same way after being kept for a minute. The 200 μL of buffer AL and 600 μL of 100% EtOH were then added to each filtrate and mixed by pipetting. The mixture was applied to a DNeasy Mini spin column and prepared according to the manufacture's manual. The sample solution (100 μL of buffer AE) was stored in a 1.5 mL microtube at −20 °C until qPCR analysis.

## Quantitative real-time PCR (qPCR)

The eDNA was measured with four PCR replicates using a PikoReal Real-Time PCR System (Thermo Scientific, Waltham, MA, USA). To detect and quantify the DNA of the red-eared slider using qPCR, the mitochondrial cytochrome b gene fragments were amplified and quantified with the following primers and probe: Tse-Kako-A-F (5′-CCTCCAACATCTCTGCTTGA -3′), Tse-Kako-A-R (5′-ATTGTACGTCTCGGGTGATG-3′), and Tse-Kako-A-MGB-P (5′-FAM- CGGAATTTTCTTGGCTATAC -MGB-3′). We used the consensus sequence of *Trachemy scripta elegans* with the following accession numbers: FJ770617, FR717131, EU787024, AF207750, and HQ442420 in NCBI (http://www.ncbi.nlm.nih.gov/). From the consensus sequence, we artificially synthesized a standard gene, which included the amplicon region and 20 bases each of the upper and lower sequences (5′- ATTCATTGATCTACCAAGCCCCTCCAACATCTCTGCTTGATGGAA

CTTTGGATCCTTATTAGGTACTTGCCTAATCCTACAAATCCTTACCGGAATTT
TCTTGGCTATACACTACTCCCCAGACATTTCACTAGCATTCTCATCAGTAG
CCCACATCACCCGAGACGTACAATACGGATGACTTATTCGTAAT-3′). The
specificity of the probe and primers was confirmed by Primer-BLAST and testing on
Japanese turtles (*Mauremys japonica*, *Mauremys reevesii*, and *Pelodiscus sinensis*). Each
TaqMan reaction contained 900 nM of each primer, 125 nM of TaqMan probe, 5 μL qPCR
master mix (TaqMan Environmental Master Mix 2.0, Thermo Scientific, Waltham, MA,
USA), 0.2 μL AmpErase® Uracil N-Glycosylase (UNG, Thermo Scientific, Waltham, MA,
USA), and 2 μL of the DNA solution. The total volume of each reaction mixture was 10
μL and we performed four replicates for PCR. The PCR conditions were as follows: 2 min
at 50 °C, 10 min at 95 °C, and 55 cycles of 15 s at 95 °C and 60 s at 60 °C. These cycles
are based on various previously described methods (*Thomsen et al., 2012a*; *Tréguier et al.,
2014*; *Katano et al., 2017*; *Doi et al., 2017b*). The qPCR results were analyzed using PikoReal
software ver. 2.2.248.601 (Thermo Fisher Scientific, Waltham, MA, USA), and the standard
curves were also automatically estimated by this software. The $R^2$ values of the standard
curves ranged from 0.960 to 0.989, and PCR efficiency ranged from 73.09 to 118.56%. For
the $R^2$ values, there is a possibility that we measured the eDNA in the qPCR system with
order-level differences because of the high variance in eDNA concentration, from 0.21 to
48,016.44 (copies/L). We defined the limit of detection (LOD) for red-eared slider DNA
as one copy per reaction based on a qPCR assay of the four replicates. Each real-time PCR
assay included four no template controls (NTCs) and we also measured the cooler and
filter blanks with four replicates. We used the average value of the four replicates for each
eDNA concentration. All of the above qPCR procedures were based on the MIQE checklist
for qPCR (*Bustin et al., 2009*). We performed the PCR set up and real-time PCR in two
separate rooms to avoid contamination.

## PCR inhibition test

We compared the Ct shift between the samples and controls with the same number of
known target DNA copies, based on the method by *Doi et al. (2017b)*, to confirm the
degree of PCR inhibition. Ct is defined as the number of cycles required for enough
amplified PCR product to accumulate that it surpasses a threshold recognized by the
real-time PCR instrumentation. The Ct is inversely related to the starting quantity of
the target DNA in a reaction and is used to calculate this quantity. The presence of
PCR inhibitors will shift (delay) the Ct for a given quantity of the template DNA. To
confirm the effects of water quality factors on eDNA inhibition, we did not use a PCR
inhibition removal kit (e.g., OneStep PCR Inhibitor Removal Kit, Zymo Research) for
the qPCR preparation. To test for PCR inhibition in the DNA samples, 1 μL of plasmid
including the cytochrome b gene from *Trachurus japonicus* ($1.5 \times 10^4$ copies), which is
a marine fish that does not inhabit the sampled ponds, was added to the PCR tempelate
with 1 μL of DNA-free distilled water. We used the primer and probe set that was
reported by *Yamamoto et al. (2016)*: forward primer: 5′-CAGATATCGCAACCGCCTTT-
3′; reverse primer: 5′-CCGATGTGAAGGTAAATGCAAA-3′; and probe: 5′-FAM-
TATGCACGCCAACGGCGCCT-TAMRA-3′. The PCR conditions were the same as

above. Each real-time PCR assay included three no template controls. We used the average value of the replicates for each Ct value. $\Delta Ct \geq 3$ cycles were considered to be evidence of inhibition (*Hartman, Coyne & Norwood, 2005*).

## Water quality analysis

We measured phosphate ($PO_4$-P), nitrate ($NO_3$-N), total phosphorus (TP), total nitrogen (TN), dissolved organic matter (DOM), and total organic matter (TOM) from the filtrate, according to the methods of *Saijo & Mitamura (1995)*, using a spectrophotometer (HITACHI U-5100, Hitachi, Tokyo, Japan). The absorbance of DOM was read at 254 nm, using samples not in an autoclave.

## SS measurement

We used the GF/F glass filter, which had been burned and dried before the weight was measured by electric balance (Sartorius CPA2252), prior to the SS analysis. The filtrate was dried in the 60 °C automatic oven (Yamato DX402, Yamato, Japan) over 12 h before the weight was measured. After that, we burned the dried filter at 450 °C for 2 h using an electric muffle furnace (Yamato FO410), then measured the weight again in the same way. The SS content was calculated as follows; (450 °C burned weight)–(60 °C dried weight).

## Chl. *a* measurement

We extracted the Chl. *a* of the filter by immersing it in 99.5% ethanol over 12 h. The extracts were measured at 630, 645, 663, and 750 nm absorbances by the spectrophotometer (HITACHI U-5100). The Chl. *a* concentration was determined according to the following equation (UNECSO 1969):

$$Chl.a \ (mgL-1) = \{(11.64 \times E663 - 2.16 \times E645 + 0.1 \times E630) \times k\}/V$$

where, k: ethanol for extraction (mL); E663, E645, and E630: each absorbance at 663, 645, 630 nm, excluding the absorbance at 750 nm; and V: water samples (L).

## Statistical analysis

We used a linear model (LM) to evaluate the relationship between the red-eared slider eDNA concentration and the number of red-eared sliders observed visually. For the eDNA concentration and the number of red-eared sliders reported by visual observation, we used the data of the sites where red-eared sliders were detected, regardless of the eDNA detection (11 ponds in which red-eared sliders were detected by both methods and 10 ponds in which they were detected only by visual observation; $N = 21$; Tables S1A, S1B). In addition, we used a multiple linear regression to evaluate the relationship between the eDNA concentration and the following environmental factors: Chl. *a*, SS, $PO_4$-P, $NO_3$-N, TP, TN, DOM, TOM, and estimated number of red-eared sliders (animals/km$^2$; from the result of eDNA concentration and the number of red-eared sliders observed visually). We used the data of all sites in which eDNA was detected (11 ponds in which red-eared sliders were detected by both methods and nine ponds in which they were detected by only eDNA, $N = 20$, Tables S1A, S1B). Prior to the multiple linear regression, we used a variance inflation factor (VIF) to check the collinearity of the factors. The maximum VIF

was 59.685, indicating that co-linearity among the factors would influence the results of the multiple linear regression. Thus, we removed the factors with a VIF >5 to reduce the collinearity effect on the multiple linear regression, resulting in the Chl. *a*, TP, and number of red-eared sliders remaining in the final linear model (LM) analysis. All statistical analysis and graphics were conducted in R ver. 3.4.1 (*R Core Team, 2018*) with the "ggplot2" and "car" packages.

## RESULTS

### The relationships between eDNA measurements and visual observations

Of the 100 surveyed ponds, we detected red-eared sliders in 30 ponds: they were detected in 11 ponds by both visual and eDNA surveys, 10 ponds by only visual survey, and 9 ponds by only the eDNA survey (Tables 1A, 1B). All successful amplifications were considered to be red-eared sliders, because the probe and primers were confirmed to detect only red-eared sliders by Primer-BLAST and were tested by real-time PCR using DNA extracted from Japanese turtles (*Mauremys japonica*, *Mauremys reevesii*, and *Pelodiscus sinensis*) living in the same region. There was a significant positive correlation between the eDNA concentrations in all ponds in which eDNA was detected and the number of red-eared sliders identified by visual observations (LM, $R^2 = 0.48$, $p < 0.001$, $N = 20$, Fig. 2).

The $\Delta$Ct values in 99 of the 100 ponds, except pond No. 33, were lower than 3 ($0.27 \pm 0.22$, mean $\pm$ SD), which means that they were lower than the inhibition criteria (*Hartman, Coyne & Norwood, 2005*). Thus, PCR inhibition was not significant for all samples, but pond No. 33 showed no amplification in the PCR inhibition test. We did not detect any amplifications in the negative controls and equipment blanks, including the cooler and filter blanks.

### The relationship between eDNA concentration and water quality

The results of the water quality analysis are shown in Tables S1A, S1B of the SEM [Chl. *a*: $0.04 \pm 0.10$ µg L$^{-1}$, SS: $35.0 \pm 44.34$ µg L$^{-1}$, PO$_4$-P: $0.78 \pm 0.87$ µmol L$^{-1}$, NO$_3$-N: $38.93 \pm 18.25$ µmol L$^{-1}$, TP: $1.00 \pm 0.99$ µmol L$^{-1}$, TN: $29.53 \pm 30.77$ µmol L$^{-1}$, DOM: $0.07 \pm 0.04$ (abs = 254 nm), and TOM: $0.00 \pm 0.01$ (abs = 254 nm), mean $\pm$ 1 SD]. The VIFs of the LM for each factor are shown in Table S2 in the SEM, of which the VIFs of the Chl. *a* (4.391), TP (1.870), and estimated number of red-eared sliders (1.801) were lower than 5. The LM, without the factors and with a VIF of >5, showed that Chl. *a* was negatively related with the eDNA concentration (LM, $p < 0.001$, Tables 2, 3), while there was no significant relationship between the eDNA concentration and other factors (Fig. 3, Tables S1A, S1B).

## DISCUSSION

We detected the eDNA of the red-eared slider in the surface water samples of the ponds. In this study, we designed the red-eared slider-specific primers and probe and showed that we can use them to detect the red-eared sliders in field samples. The red-eared slider is a

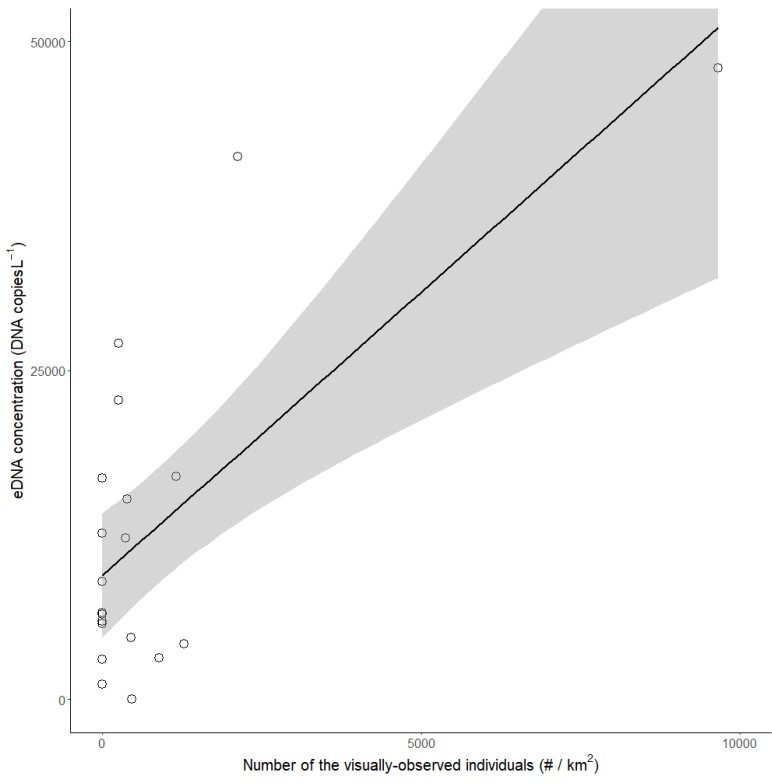

**Figure 2** **Relationship between the visual observation number of red-eared sliders (*Trachemys scripta elegans*) per km² and their eDNA concentrations in the ponds.** The grey area represents the limits of the 95% confidence interval for the slope of the linear regression ($R^2 = 0.48$, $p < 0.001$).

common invasive species in Japan, but we found them in only a few sites. This is probably because we randomly selected the study ponds from each region, including less invasive sites in the mountain area. When comparing with the visual observations, we detected the red-eared slider by both the eDNA and visual surveys in 11 ponds. In the nine ponds where only eDNA detected turtles, we suspect that turtles were too rare to be observed by visual observation. In addition, eDNA methods have often been cited to detect rare or cryptic species (*Barnes & Turner, 2016*). Our results in nine ponds support this phenomenon as the turtles were detected not by visual observation, but by eDNA. While, in the ten ponds detected by only visual observation, we sampled the eDNA at one point per pond as per the survey design. This result suggests that it was necessary to sample at several points at each study site, to decrease the false-negative eDNA detections (e.g., *Tréguier et al., 2014*; *Thomsen et al., 2012a*). It would be necessary to compare eDNA sampling at several points at each study site with other surveys in future studies. In summary, we can detect red-eared sliders using eDNA for two thirds of the total detections, by sampling only 500 mL of water at a point in the pond. However, we concluded that it might be appropriate to use both a visual survey and eDNA analysis for red-eared sliders, as eDNA produced false-negative detections when red-eared sliders were present.

**Table 2** Linear regression slopes with a ±95% confidence interval, SE, $t$ values, and $p$ values for the relationships between Chl. $a$, TP, and eDNA concentrations in the ponds. Factors with a VIF > 5 were removed.

(A)

| Factors | Slope | SE | $t$ value | $p$ value |
|---|---|---|---|---|
| Chl. $a$ | −48.550 | 10.740 | −4.519 | 0.000 |
| TP | −0.345 | 0.392 | −0.882 | 0.393 |
| Turtles | 0.000 | 0.000 | 2.075 | 0.057 |
| Intercept | 10.450 | 0.754 | 13.865 | 0.000 |

(B)

| N | 18 |
|---|---|
| $F$ value | 12.73 |
| $p$ value | 0.000 |
| $R^2$ | 0.732 |
| Adjusted $R^2$ | 0.674 |

**Table 3** The table represents the $n$, $F$ value, $p$ value of the $F$ value, $R^2$, and adjusted $R^2$ for the linear regression.

| | |
|---|---|
| $F$ value | 12.73 |
| $p$ value | 0.000 |
| $R^2$ | 0.732 |
| Adjusted $R^2$ | 0.674 |

We also found a significant positive correlation between the eDNA concentration and number of red-eared sliders detected by visual observation. Others, such as *Takahara et al. (2012)* have found positive correlations between eDNA concentrations and the biomass using both aquaria and outdoor experiments. The field tests for the relationship between eDNA concentration and biomass of amphibians and fish also support the positive correlation (e.g., *Pilliod et al., 2013*; *Doi et al., 2017b*). Although few studies examining the effects of biomass or number of individuals on eDNA have used reptiles, *Lacoursière-Roussel et al. (2016)* found eDNA detection rate was highly correlated to the relative abundance of wood turtles. In our study, likewise, a positive relationship between the eDNA concentration and the abundance of visual detections was observed. Thus, we can possibly use the eDNA concentration to estimate the number of red-eared sliders, especially for the ponds with a high species abundance.

From the relationships between water quality and eDNA concentration, Chl. $a$ seemed to influence the degradation of eDNA. One of our hypotheses was that water quality would influence/inhibit qPCR for eDNA detection, however, we could not find any inhibition of PCR by inhibition tests, except for a single pond (No. 33). Only pond No. 33 could be inhibited, because it showed no amplification of DNA. It is considered that one of the inhibitors of the PCR for pond No. 33 was humic acid from the decomposition of leaves, which is known to inhibit PCR detection (*Opel, Chung & McCord, 2009*), because it was surrounded by forest and the water color was black (Fig. S2 in SEM). Thus, Chl. $a$

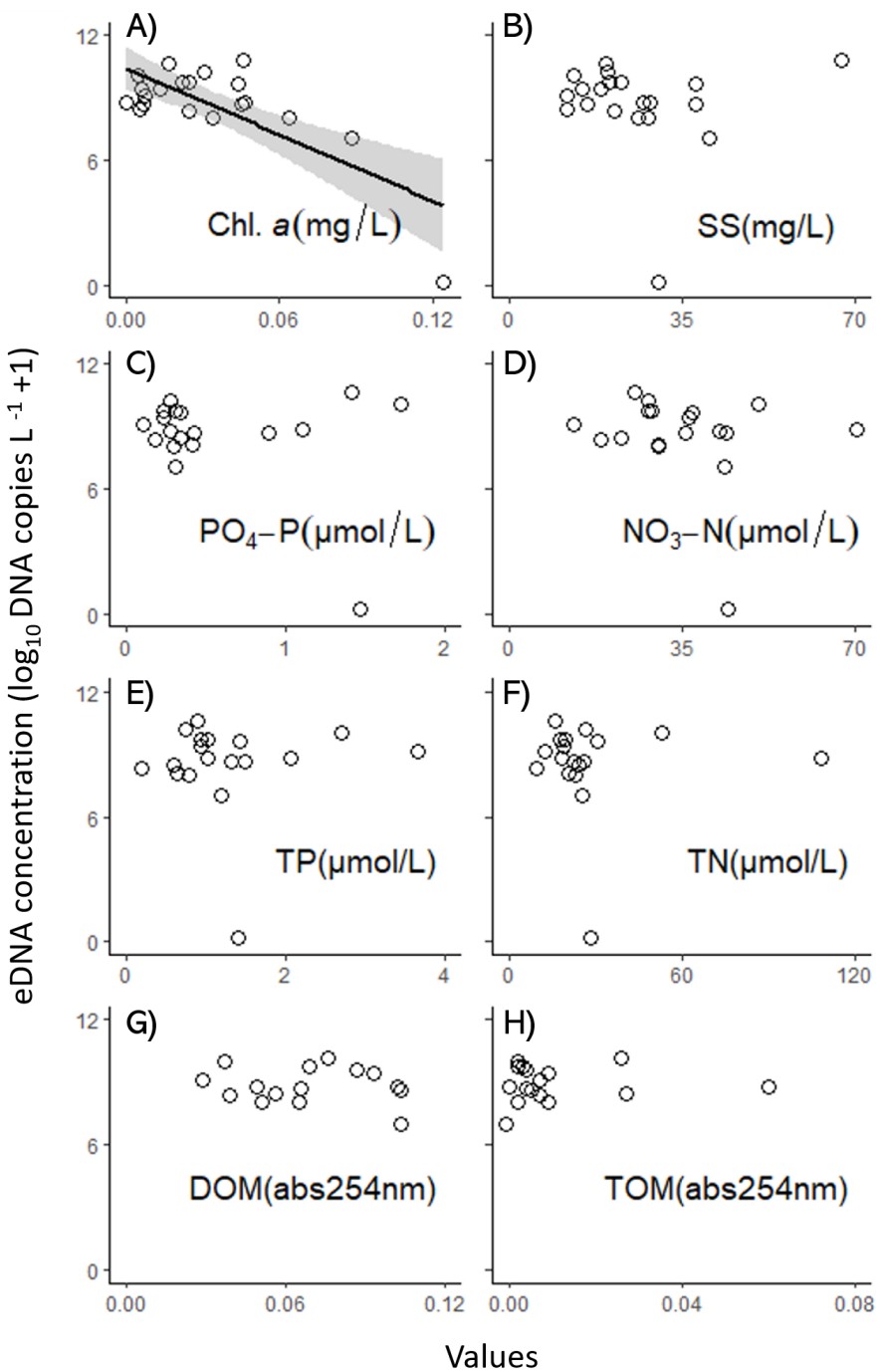

**Figure 3  Relationships between each water-quality factor and the eDNA concentration of the red-eared slider in the ponds (A: Chl. *a*, B: SS, C: PO $_4$-P, D: NO₃-N, E: TP, F: TN, G: DOM, H: TOM).** The regression curve of Chl. *a* was drawn by linear regression with 95% confidence intervals for the slope ($R^2 = 0.54$, $p < 0.001$).

might not be directly related to PCR inhibition; however, it might have an influence with respect to decreasing the eDNA concentration in the water through DNA degradation. For the other water quality characteristics in our study, *Eichmiller, Best & Sorensen (2016b)* measured the effects of Chl. *a*, TN, TP, and SS on the decay rate of carp eDNA in laboratory experiments, however, these variables were not significantly correlated with the eDNA decay rate (*Eichmiller, Best & Sorensen, 2016b*). This result of non-correlations with TN, TP, and SS to eDNA degradation was the same as our results, however, we showed the negative relationship with Chl. *a* and eDNA.

Our result that only Chl. *a* had a significant effect, might suggest that Chl. *a* influences the degradation of eDNA in the surface water of ponds. However, this phenomenon seems to be in debate. For example, the eDNA decay rate has a negative relationship with Chl. *a* in a mesocosm experiment for goldfish (*Barnes et al., 2014*), i.e., the eDNA degradation was less in the higher Chl. *a*. On the other hand, the eDNA detection rate has a negative relationship with Chl. *a* in a field survey for silver carp (*Song, Small & Casman, 2017*). In our study, the eDNA concentrations have a negative relationship with Chl. *a*, which supports the results of (*Song, Small & Casman, 2017*). As abiotic environmental factors indirectly influence the increase of microbial activities, eDNA may be decomposed by microorganisms (*Barnes et al., 2014*). Thus, the eDNA degradation by microorganisms bonded to phytoplankton, for example, indirectly increases microbial activities by providing basal resources (*Lennon, 2007*), although we did not directly evaluate the microbial activity. Further discussion on the "Chl. *a* hypothesis" on eDNA degradation is required for understanding the mechanisms of eDNA degradation and for developing eDNA methods, especially for eDNA surveys in highly-productive water bodies. In this study, we can provide the hypothesis from the field data, but further field and laboratory experiments controlling the DNA concentration and water conditions, including the water quality and planktonic community, are required for understanding the mechanisms. In addition, we detected eDNA in only 20 ponds (11 ponds by both eDNA and visual observation and 9 ponds by only eDNA) by a single sampling from each pond. In future field studies, more replications would be needed.

## CONCLUSION

In conclusion, we detected red-eared sliders using eDNA at a similar performance to that using visual observations, and evaluated the abundance using the eDNA concentration obtained from a single sampling point at each pond. In future field studies, more replications at each sites would be needed, as other studies (e.g., *Barnes & Turner, 2016*) reported, which suggested that multiple water sampling points at a pond would increase the ability to evaluate the distribution and abundance by eDNA. Likewise, to prevent the spread of red-eared sliders, further developments of the eDNA method to easily detect them will be necessary in the future. We also provide the "Chl. *a* hypothesis" for eDNA degradation for comparing the water quality of the ponds. For eDNA surveys, we should pay attention to the potential for false-negative detections, probably because of the state of primary production with reference to the Chl. *a* concentration. Understanding the mechanisms in eDNA degradation would provide us with the tools for easy and accurate eDNA methods to evaluate the distribution of aquatic organisms.

## ACKNOWLEDGEMENTS

We thank A. Sumi and D. Togaki for their helps on our sampling and experiments.

### Funding

This study was supported by the Environment Research and Technology Development Fund (4-1602) of the Ministry of the Environment, Japan and JST-CREST (JPMJCR13A2) for Hideyuki Doi and Toshifumi Minamoto and JSPS KAKENHI (15K00596, 15K07233) for Izumi Katano and Toshifumi Minamoto, respectively. The funders had no role in study design, data collection and analysis, decision to publish, or preparation of the manuscript.

### Competing Interests

Rio Souma is an employee of IDEA Consultants Inc.

### Author Contributions

- Aozora Kakuda conceived and designed the experiments, performed the experiments, analyzed the data, prepared figures and/or tables, authored or reviewed drafts of the paper, approved the final draft.
- Hideyuki Doi performed the experiments, analyzed the data, authored or reviewed drafts of the paper, approved the final draft.
- Rio Souma and Mariko Nagano performed the experiments, authored or reviewed drafts of the paper, approved the final draft.
- Toshifumi Minamoto performed the experiments, contributed reagents/materials/analysis tools, authored or reviewed drafts of the paper, approved the final draft.
- Izumi Katano conceived and designed the experiments, performed the experiments, analyzed the data, authored or reviewed drafts of the paper, approved the final draft.

### Data Availability

The raw data is available as a Supplemental File.

### Supplemental Information

Supplemental information for this article can be found online at http://dx.doi.org/10.7717/peerj.8155#supplemental-information.

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
