# Peer review of "Environmental DNA detection and quantification of invasive red-eared sliders, Trachemy scripta elegans, in ponds and the influence of water quality"

_PeerJ, doi:10.7717/peerj.8155_

## Round 0.1 · original submission · Major Revisions

Thank you for your submission. Please find two detailed and well crafted reviews to help strengthen and revise your manuscript. It is clear from the reviews, this paper is worthy of publication. Please make a full faith effort to work through these reviews and take advantage of their detail. I look forward to seeing your revision.

Reviewer 1 ·

Basic reporting

Overall, I think that the authors are direct to their points and the introduction is well-written. I have suggested some corrections and made more specific comments below.

L44-47: I might suggest that you flip those sentences, so you start by explaining what eDNA is and then transition into how it can successfully be used to detect a variety of aquatic organisms. Additionally, as Peer J is an open-source journal, some readers may not know exactly what eDNA is or the capabilities of eDNA.

L45-46: You use the reference style of “Goldburg and Strickler, 2015”, but at L50: you use the style of “Thomsen & Willerslev 2015". Please double-check citation style throughout.

L47: You do not always need to filter litres of water to detect an organisms’ eDNA. I would suggest that you revise to say something about how you can filter eDNA from water and detect organisms without necessarily needing to directly observe them. As this is one of the most compelling part of eDNA, that it can often detect low-abundance organisms.

L55: Uncertain if this should be “trematode” or “trematodes”?

L56-57: I would argue that we cannot always find relationships between the amount of eDNA and the biomass or abundance. Some studies can, some cannot. In my opinion, the fact that here you claim to find that relationship is one of the key findings of this paper (although see below for additional comments on this claim).

L64: Please add a reference here.

L66-68: I think a reference is necessary here. I could not seem to find this reference cited in your “References” section.

L68-71: I think a reference or references are necessary here.

L87-88: This sentence is worded awkwardly and I believe there are other inhibitors other than humic acid. Perhaps, you could revise to read something like, “For example, humic acids are known to inhibit DNA polymerase in PCR and therefore could directly impact eDNA detection.”

L100: I would suggest to delete the “ ,” after (qPCR) and say something like “paired with visual observations of turtles.”

Experimental design

L101: 100 ponds! That’s quite a robust sample size! But then later the number of detections is concerning. If red-eared sliders are a common invasive species in Japan, please add in your discussion more about why you did not detect more turtles.

L118-119: Uncertain exactly what “From a point within each pond, …” means. Please explain how water sampling locations were selected more. Randomly? As water/conditions allowed?

Why did you chose to limit your sample size to only 1, 500mL bottle per pond? That seems very low replication.

L153-155: Please provide the ascension number from GenBank or some additionally information on what you used as your positive turtle standard. This will be important for other seeking to use this assay in the future.

L164: 55 cycles is a long time. Please provide additional information on why the qPCR methodology is so long. It’s not necessary unheard of, but without information about detection levels (see comment below), I don’t know why 55 vs. 40?

L:166-167: Please provide more information about your limit of detection (LOD) and limit of quantification (LOQ). It is not stated here how low your assay was capable of detecting red-eared slider DNA. This limits the ability of future eDNA researchers to use this assay.
Additionally, your R2 values and %efficiency are low and I’m concerned about your ability to accurately measure eDNA concentration. Please provide additional information or references to support this. For example, see Hinlo et al. 2018 (Performance of eDNA assays to detect and quantify an elusive benthic fish in upland streams. Biological Invasions 20:3079-3093.) examined quantified eDNA, but reported R2 values of 0.999 and %efficiency of 96.908.

L174: Unclear why a PCR inhibition removal kit (e.g., Zymo Research, OneStep PCR Inhibitor Removal Kit) was not used on your same samples to examine if you directly had inhibition in your turtle eDNA that could influence your results. Please explain.

L218-220: Please explain why you used linear models and not linear mixed-effect modeling with site a random variable? Then use AICc model selection on only the variables that are not co-linear?

L232: Did you sequence confirm that the successful amplifications were in fact red-eared slider? Even just one or two eDNA samples to make sure that your assay works (you tested on cyt b, but what about your positive eDNA detections?). Sometimes and even with the specificity of a probe, you can still amplify non-target organisms. Please clarify how you are certain that positive detections are not "false-negatives" from primer-dimer.

Do the authors conclude that 1 amplification by Ct 55 is an amplification and the take the mean of that? Please clarify.

Validity of the findings

The authors did an excellent job in the transparency of their data-a lot of information in contained in the Tables and Figures.

However, I have serious concern over the authors’ abilities to correctly determine eDNA concentrations (see above about R2 and % efficiencies). Further, I would suggest that this discussion needs to be revised to discuss the authors’ shortcomings. I don't agree that not-finding a significant result or no trend means that this paper is unworthy of publication. Instead, I suggest that others are more straight-forward in not making large conclusions from small data and that they should discuss what future research could be done.

L235: 100 lakes sampled and being able to only determine red-eared sliders in 11 ponds, I am uncertain how your N=20. If 10 ponds detected with visual, 9 with eDNA, I’m confused how that is 20.

L231-235: I would be hesitant to infer much from detection in only 10 ponds.

L257-259: I disagree that you can make that conclusion based on only being able to detect eDNA at 9 ponds. Your results actually indicate that you’re able to detect the turtles better with visual observation.

Please revise this to discuss more about the short-comings of your paper and how future research can add to your study.

Additionally, you discuss in the introduction about how visual surveys are not the best method to detect these turtles. Maybe eDNA sampling at one point vs. visual surveys is not the comparison that is needed and in the future, maybe more work should be done to compare other samplings methods vs. eDNA (at with more than n=1 at each of your ponds).

L295: I believe this should be “Barnes et al. 2014”?

L292-308: The relationship between Chl a and eDNA is interesting, however, I would suggest that you again state that eDNA was only detected in 9 ponds.

Tables/Figures:

Table S1: Please explain why some values are missing. Were these not measured? Or were they removed from analyses?

Figure 2: The eDNA concentration is not log-transformed here, but is in Figure 3? Please explain that, as I do not see a trend here that you are concluding. Instead this relationship seems to be largely driven by the point of 10,000 red-eared turtles. Further, were you able to count 10,000 turtles?

Figure 3: I would suggest that you add regression curve + 95% CI to all graphs. I understand that only Chl. a had a significant effect on eDNA copy number, but it looks odd to not show the other trendlines.

Figures 2 and 3: Suggest to add r2 values and p-values to figures?

Additional comments

It is very exciting to see a turtle eDNA paper, as reptile eDNA is understudied! Further, more studies that directly examine the relationship between number and eDNA are necessary, as there do appear to conflicting results dependent on organism and system studied.

However, I have serious issue with the overstatement of conclusions that the authors report here, largely due to the small number of ponds where turtles were actually collected. I am also hesitation to conclude much give the low R2 values and wide-range in % efficiency. I would suggest that the authors specifically address those concerns, as this assay and study does present interesting findings that other researchers would be very interested in reading!

·

Basic reporting

Literature references, article structure, figures, tables, and data sharing are all sufficient. The English needs clarification or correction in some cases (many of which I have pointed out in the General Comments). Hypotheses are not explicitly stated, but could easily be added to the last paragraph of the Introduction.

Experimental design

Methods are generally well-described, but a few details need to be added as noted in the General Comments section. One statistical analysis, the testing of eDNA concentration against water quality metrics, is confounded by varying slider turtle abundances. I have provided suggestions for overcoming this issue in the General Comments section.

Validity of the findings

Conclusions about the effect of water quality metrics on eDNA concentrations are based on a confounded analysis as mentioned above and in the General Comments. This major issue needs to be corrected before any conclusions can be made. The other conclusions in the manuscript are fine.

Additional comments

The manuscript "Detection of environmental DNA of the invasive red-eared slider in ponds for evaluating their distribution with comparison to water quality" by Kakuda et al. presents the results of a survey of turtle presence/abundance and water quality in 100 ponds in Japan. The authors compare visual versus environmental DNA (eDNA) survey results, test eDNA samples for PCR inhibition, and test the relationship between eDNA concentration and various water quality parameters. The authors find evidence for slide turtles in 30 out of 100 ponds, with 10 ponds having just visual detections, 9 ponds having just eDNA detections, and 11 ponds having both visual and eDNA detections. Only one pond had evidence for PCR inhibition, which was surprising to me. Finally, the only significant relationship between eDNA concentration and water quality was a negative relationship with Chlorophyll a. The authors conclude that eDNA is as good as visual observation for slider surveys, and that Chl. a may increase DNA degradation rates in the field, but more research is needed for the latter.

This is a very interesting study and provides valuable data the increases our understanding of the field “ecology” of eDNA. The introduction includes relevant literature citations. The methods, while sometimes needing more details (see Minor Comments below), are sound. My two major issues with the manuscript are 1) the writing, which is sometimes unclear (see Minor Comments below for some suggestions), and 2), more importantly, the eDNA concentration/water quality analysis. For the latter, the authors use linear regression to test relationships between eDNA concentration and various water quality parameters. However, they neglect to include slider turtle abundance into this analysis, even though they clearly show that slide turtle abundance as measured visually is strongly correlated with eDNA concentration (Lines 233-235, R2=0.48, p<0.001). It could be that the negative relationship that they find between eDNA concentration and Chl. a is entirely due to a negative relationship between slider turtles and Chl. a. To deal with this confounding issue, the authors could include visually-estimated slider turtle abundance data as a covaraiate, but this would limit them to just 11 data points, which I'm not sure provides enough statistical power. They may be able to add 9 more data points by assuming low abundance in the eDNA detection-only sites. But a new analysis that considers slider turtle abundance is clearly needed in order to properly evaluate the Chl. a result.

Below I note some minor comments that I hope will improve the readability f the manuscript and replicability of the study. I have not made suggestions for the Discussion section as I think that needs to be completely re-written after a new eDNA concentration ~ water quality analysis is done.

Minor Comments

Title
This is a confusing and long title. I suggest something like “Environmental DNA detection and quantification of invasive red-eared sliders, Trachemy scripta elegans, in ponds and the influence of water quality”. The scientific name for red-eared sliders needs to be included in the title.

Abstract
L26. Environmental effects on eDNA don't necessarily prevent accurate estimates, but they do complicated estimates. I would change “prevent” to “complicate”.
L28: Change “Also” to “Further”
L29-32: Suggest rewrite “Here we examined the relationship between the visually-observed number of red-eared sliders (Trachemys scripta elegans) and eDNA concentrations across 100 ponds. Additionally, we evaluated the effect of water quality on slider eDNA concentration in these ponds.”

Introduction
L46: Change to “Environmental DNA consists of DNA fragments...”
L47: Some eDNA surveys use only 250ml, so change to “a few liters or less...”
L49: Change short to “shorter”.
L56-57: Delete sentence “We can detect the eDNA of various taxa...” because it's redundant with 1st sentence.
L60: Change “detect” to “determine”.
L67: The statements “which was 64% of all turtle abundance” is confusing – were red-eared sliders 64% of all invidual turtles observed?
L76: Change “the previous eDNA studies” to simply “previous eDNA studies” (delete “the”)
L87: Rewrite “In fact..” sentence because it is awkward. I suggest “Inhibition of eDNA detection can be caused by compounds such as humic acids that inihibit the DNA polymerase used in PCR.
L90-94: Mention that both the Strickler et al. and Barnes et al. studies are mesoscosms earlier. I suggest: “Mesocosm studies have found that eDNA degradation decreases with low water temperatres, low UV-B levels, and high PH (Strickler et al. 2014) and with increasing biochemical oxygen demand, Chlorophyll a concentration, and total eDNA concentration (Barnes et al., 2014).”
L95: Change “conclude on” to “understand”
Line 100: Change “Additionally, we aimed to” to “Specifically, we evaluated”.
*Also, you should include a hypothesis/es in the last paragraph of the Introduction

Methods
L109: How/why were these ponds chosen? Also, you talk about city, rural, and mountain areas but do not consider these in the analysis or the discussion. Were there any interesting patterns amond these three area types?
L118: What time of day were visual surveys done? Was weather considered as a cofactor, since turtle bask more on sunny days?
L128: Would be helpful to specify again that 500mL was filtered for eDNA collection.
L132: Suggest rewrite sentence to “For eDNA samples, the 'cooler blank' and a 'filter blank' consisting of DNA-free distilled water were filtered in the same was as the samples.”
L137-138: The statement “in the 100 samples without six data is confusing”. I suggest changing to “data missing from six of the 100 sites”. Also, from the table is looks like more than six sites are missing water quality data, so please check this number.
L166: Please describe how you estimated your standard curves.
L181-182: Suggest rewrite “To test for PCR inhibition in the DNA samples, 1μl of plasmid including the cytochrome b gene from Trachurus japonicus..”. Also, what did you mean by “1.5 X 104 copies”, why not say 156 copies?
L226: What R package was used for the VIF analysis? Need to include that.

Results
L232: Delete “Of the 11 ponds...” because this suggests that the 10 visual-only and 9 eDNA-only ponds were included in the 11 visual + eDNA detection ponds. Rewrite these first two sentences to something like “Of 100 samples ponds, we detected red-eared sliders in 11 ponds with visual and eDNA surveys, in 10 ponds with just the visual survey, and in 9 ponds with just the eDNA survey (Table 1).”

---

## Round 0.2 · Minor Revisions

Well done! Thank you for your revision and attention to detail in improving the manuscript. Based on two re-reviews, this paper is nearly ready for publication. Please work through the minor comments from the two reviewers. With minor revision, this paper will be accepted in PeerJ. Thank you!

Reviewer 1 ·

Basic reporting

Some minor issues with English grammar that can easily be addressed by the authors.

Experimental design

No comment-
Issues with sampling design (one sample/pond) are stated as downfalls in the discussion section.

Validity of the findings

No comment-
Findings confirm previous research and present novel results on the relationship between Chl a and eDNA.

Additional comments

It seems that the authors have taken appropriate steps in revising the manuscript to address the concerns of the reviewers. Overall, the paper is generally well-written and would likely be of interest to the eDNA community.

I suggest minor corrections to the English language to clarify grammar and assist with sentence flow.
L39: I suggest "inhibition" not "inhibitions"

L42: Suggest stating, "...the ponds, as microbial activity is thought to decrease eDNA persistence." It was not immediately clear what was being stated here.

L95: Suggest revision to read something like, "Mesocosm studies have found that a number of factors are important to decreasing the degradation of eDNA, such as cooler temperatures, higher BOD, increased Chl a, (Barnes et al., 2014; Strickler et al., 2014)."

L99: Suggest revision to say, "Because the effect of Chl a was different in laboratory vs. in situ experiments, more evidence is needed to understand the effects of water quality on eDNA degradation."

L 101-103: Can delete? This is repetitive to sentence before.

L196: One copy/µL?

L240: Revise to "...were measured at 630..."

L305-308: Suggest revision for clarity to be something like, "In the nine ponds where only eDNA detected turtles, we suspect that turtles were too rare to be observed by visual observation. In addition, eDNA methods have often been cited to detect rare and cryptic species (Barnes & Turner, 2016)." Then delete the next sentence as it is repetitive (Lines 307-308).

L310: Add "s" to suggest

L313: Delete this empty space and bring the last sentence of the paragraph up to line 313.

L316-317: Suggest revision to read something like, ", as eDNA produced false, non-detections when red-eared sliders were present."
L319-321: Revise to, "Others, such as Takahara et al., (2012) have found positive correlations between eDNA concentrations and the biomass using both aquaria and outdoor experiments."

L322: Delete "ed" from supported to read "support"

L323-324: Suggest rearranging sentence to read, "Although few studies examining the effects of biomass or density on eDNA have used reptiles, Lacoursière-Roussel et al. (2016) found eDNA was highly correlated to the relative abundance of wood turtles. Then I suggest deleting the next sentence (L325-327) as this information has already been stated in the above sentences.

L327: Suggest replacing "for estimating" to "to estimate"

L334: Inconsistency in how pond No. 33 is labeled, see Line 332. Revise to be same format.

Line 335 and paragraphs on Chl a. Do you think that the dark tannin colored water is just less influenced by UV-B rays and temperature was cooler (unfortunately there is no measure of temperature on this) and thus, eDNA persists longer? Not necessarily a factor of microbial activity?

L346-349: The authors did a nice job of pointing out that differences between laboratory vs. in situ eDNA studies often find contradictory results.

L367-369: I disagree with this sentence, as you state on Line 360-362 that eDNA was only detected in 11 ponds more replicate water samples would be useful. In addition, you report at L254 and L260 that visual surveys detected turtles in 10 of the ponds and eDNA was only successful in detecting turtles in 9.

L372-374: Suggest delete sentence on "For eDNA surveys,...", as it has already been stated above.

L 445, 463: Trachemys scripta should be italicized. L496: italicize Cyprinus carpio

Table: Suggest that Table 2 B be a standalone table.

·

Basic reporting

no comment

Experimental design

no comment

Validity of the findings

The only criticism I have is the statement on Line 367-368 ("This result contrasts with other studies... which suggested that multiple sampling points would increase the ability to evaluate distribution and abundance..."). These results do not contrast with these studies, since you did not compare multiple versus single samples per site. I would guess that multiple samples per site would increase your ability to detect T.s. elegans, but this was not tested in this study. I would reword this sentence accordingly.

Additional comments

no comment

---

## Round 0.3 · accepted · Accept

Thank you for your efforts and congratulations on a job well done. You have been thorough and professional in your effort to revise this paper and it reads well.